# ADAMEM: Test-Time Adaptive Memory for Language Agents

**Yunxiang Zhang** [1]   **Yiheng Li** [1]   **Ali Payani** [2]   **Lu Wang** [1]

## Abstract

A central challenge for language agents is utilizing past experience to adapt to dynamic test-time conditions. While recent work demonstrates the promise of agentic memory mechanisms, most systems restrict retrieval to episode initiation. Consequently, agents are forced to rely on static guidance that becomes increasingly misaligned as long-horizon tasks unfold. To address this rigidity, we propose the **Adaptive Memory Agent (ADAMEM)**, a novel framework for agent test-time adaptation. Without updating model parameters online, ADAMEM adapts agent behavior via a hybrid memory architecture: it maintains a *long-term trajectory memory* of raw experiences collected offline while generating dynamic *short-term strategy memory* on-the-fly to guide decision-making. This mechanism enables the trade-off between token efficiency and adaptability across varying inference-time compute levels. Empirically, ADAMEM significantly outperforms static memory baselines, achieving relative gains of up to 13% on ALFWorld and 11% on WebShop, with consistent leading performance extending to agentic search on HotpotQA. To further enhance this adaptation, we develop **STEP-MFT**, a **Step**-wise **M**emory **F**ine-**T**uning technique that trains the policy to synthesize high-quality strategies from retrieved experiences, yielding additional performance gains. Our work establishes a new scaling dimension for agentic memory, supporting continuous reasoning and self-evolution post-deployment in real-world environments. Our code is available at https://github.com/yunx-z/AdaMEM.

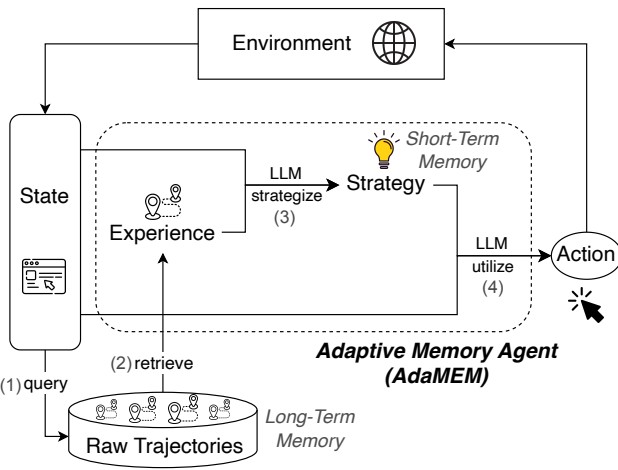

*Figure 1.* Overview of ADAMEM. Instead of relying on a static, episode-level strategy, the agent adapts to the current decision step by querying a *long-term trajectory memory* of raw experiences (1-2) and synthesizing them into a dynamic *short-term strategy memory* tailored to the current *state* (3). Conditioned on this *test-time* strategy, the agent adapts its next action (4) without requiring parameter updates.

## 1. Introduction

Building agents that can learn from experience and adapt to new situations is a longstanding aspiration in artificial intelligence (Franklin & Graesser, 1996; Silver & Sutton, 2025). With recent advances in large language models (LLMs), agents can now address long-horizon tasks including web navigation (Qi et al., 2025; Khalifa et al., 2026), coding (Jimenez et al., 2024; Zou et al., 2025) and scientific discovery (Chen et al., 2024; Jansen et al., 2024; Zhang et al., 2025a; 2026b). Despite this progress, they still struggle with **test-time adaptation** (Xiao & Snoek, 2024), the ability to effectively adjust strategies in dynamic environments during inference. While this is often addressed via test-time training (Sun et al., 2020; 2023) in machine learning and computer vision, updating model parameters for each instance or step is prohibitively expensive for long-horizon settings. Consequently, recent work has prioritized training-free prompt adaptation (Suzgun et al., 2025a), facilitated by agent memory mechanisms (Hu et al., 2026). Existing approaches typically augment the system prompt

---

[1]University of Michigan [2]Cisco Research. Correspondence to: Yunxiang Zhang <yunxiang@umich.edu>.

*Proceedings of the 43rd International Conference on Machine Learning*, Seoul, South Korea. PMLR 306, 2026. Copyright 2026 by the author(s).

with a retrieved trajectory (Zheng et al., 2024a) or a distilled strategy (Ouyang et al., 2025). Yet, these systems generally restrict memory retrieval to occur only *once* at episode initiation.[1] This creates a bottleneck of *rigidity*: by front-loading retrieval, agents are forced to rely on static guidance in the system prompt that cannot adapt to intermediate failures or shifting sub-goals. This limitation motivates our primary research questions:

1. *How can agent memory overcome the rigidity of static retrieval to adapt continuously during inference?*

2. *How can we efficiently train them to synthesize strategies that explicitly optimize this test-time adaptation?*

To address the first research question, we propose the **Adaptive Memory Agent (ADAMEM)**, a framework that enables *step-wise* test-time adaptation for language agents. As shown in Figure 1, ADAMEM decouples memory storage from abstraction through a hybrid architecture. We maintain a scalable *long-term trajectory memory* of raw experiences stored offline. At inference time, rather than retrieving raw logs, ADAMEM dynamically generates a concise *short-term strategy memory* tailored to inform the agent reasoning at specific steps.

This framework allows practitioners to optimize the trade-off between token efficiency and adaptability. We introduce flexible inference modes, ranging from ADAMEM-LOW, which prioritizes efficiency by refreshing strategies only when necessary, to ADAMEM-HIGH, which maximizes adaptability via step-wise regeneration. Empirically, ADAMEM achieves relative gains of up to 13% on ALF-World (Shridhar et al., 2021) and 11% on WebShop (Yao et al., 2022) upon static memory baselines, with consistent leading performance extending to agentic search on Hot-potQA (Yang et al., 2018). By decoupling storage from abstraction, ADAMEM allows the policy model to utilize *off-policy* long-term memory constructed by another model while ensuring the short-term strategy remains strictly *on-policy*, thus enabling *cross-model generalization*.

Despite the effectiveness of this training-free approach, simple prompting often yields over-general strategies that lack the concrete diagnoses required to recover from specific failures mid-exploration (Jiang et al., 2025a). To bridge this gap, we must explicitly train the model to generate superior strategies that actively drive decision-making. However, standard outcome-based supervision is noisy, indistinguishably crediting both crucial insights and redundant text (Jin

et al., 2025; Wei et al., 2025). To achieve fine-grained credit assignment without the prohibitive cost of auxiliary rollouts or specialized critics (Wang et al., 2024b; Chae et al., 2025), we introduce **Step**-wise **M**emory **F**ine-**T**uning (**STEP-MFT**). This critic-free method filters strategies for supervised fine-tuning based on a simple intuition: a valid strategy must alter the agent action to steer the trajectory toward success. This efficiently trains a more capable agent that outperforms both prompt-only and outcome-based counterparts without heavy computational overhead.

Our contributions are summarized as follows:

- We propose **ADAMEM**, a hybrid memory framework that enables continuous test-time adaptation by dynamically synthesizing context-aware strategies instead of retrieving static artifacts.
- We introduce **STEP-MFT**, a rejection sampling fine-tuning technique that trains agents to generate high-utility strategies using *process-level rewards*.
- We demonstrate that ADAMEM establishes a new **scaling dimension for agentic memory**. Both our training-free and STEP-MFT variants achieve a superior Pareto frontier between inference cost and task performance, yielding relative gains of up to 17% on ALFWorld and 13% on WebShop compared to static memory baselines.

## 2. Related Work

Memory serves as the cornerstone of long-horizon reasoning by enabling agents to ground current decisions in past experiences (Hu et al., 2026; Jia et al., 2026; Luo et al., 2026), yet current systems predominantly rely on retrieving static artifacts to guide inference. These range from raw trajectories (Zheng et al., 2024a; Zhao et al., 2024; Zhou et al., 2025) and distilled insights (Ye et al., 2025; Huang et al., 2025) to structured workflows (Wang et al., 2025; Tang et al., 2025b) and schematic reasoning patterns (Yang et al., 2024b; Ouyang et al., 2025; Kim et al., 2025). Skill-based memory mechanisms (Wang et al., 2024a; Zheng et al., 2025; Fang et al., 2026; Han et al., 2025) extend this by actively expanding the agent's action space via executable code or procedural modules. Yet they all share a fundamental reliance on *pre*-computation where the memory is constructed before the current task instance begins. This enforces a rigid dependency on static priors matching evolving states. We overcome this by dynamically synthesizing fresh memory online, ensuring robustness to test-time distribution shifts.

Recent work on hybrid memory architectures have sought to automate memory maintenance via dynamic optimization. MemRL (Zhang et al., 2026a) and Agentic Memory (Yu et al., 2026) leverage reinforcement learning to estimate memory utility or learn management policies at runtime.

---

[1]Following the taxonomy in Hu et al. (2026), we use "agent memory" in this work to refer specifically to *experiential memory* (procedural knowledge abstracted from past trajectories for multi-step tasks), distinct from *factual memory* (declarative knowledge for multi-session chat) or *working memory* (active context management).

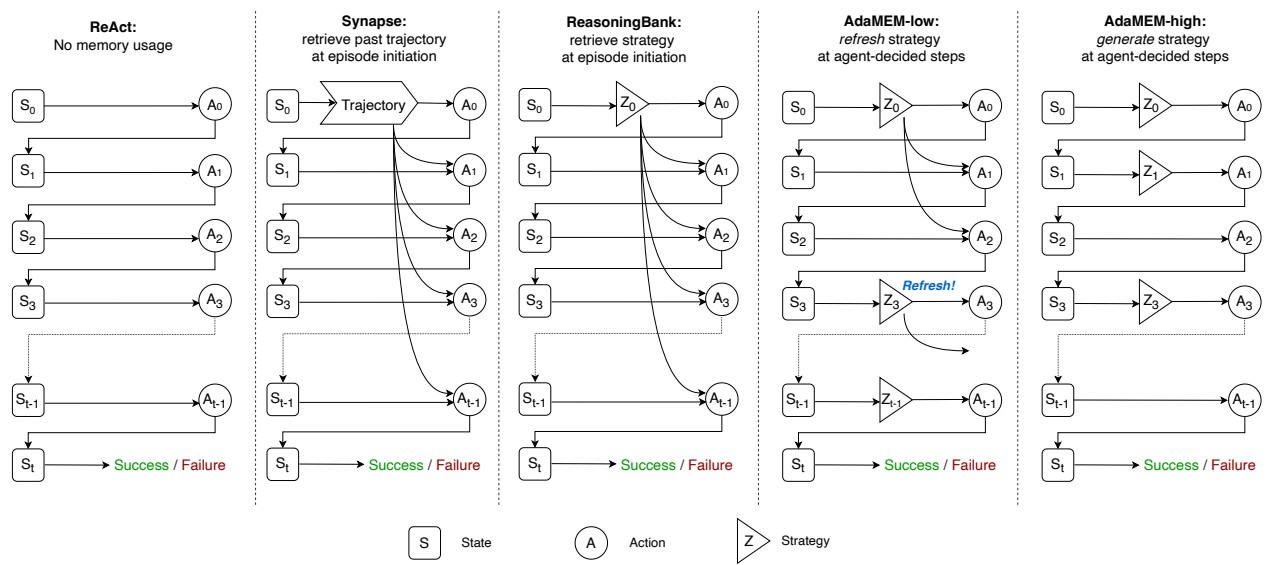

*Figure 2.* Comparison of test-time agent memory mechanisms. **ReAct** operates without external memory. **Synapse** and **ReasoningBank** employ *static initialization*, retrieving a trajectory or strategy ($Z_0$) only at the episode start ($S_0$), which forces the agent to rely on fixed guidance throughout the long-horizon task. In contrast, **ADAMEM** enables *test-time adaptation* via dynamic memory retrieval and synthesis. **ADAMEM-LOW** balances efficiency by maintaining a persistent strategy, triggering a *refresh* only when necessary (e.g., at $S_3$). **ADAMEM-HIGH** maximizes adaptability by regenerating a fresh strategy ($Z_t$) at every critical decision step. This flexibility allows ADAMEM to actively adjust its reasoning to evolving task states.

Domain-specific architectures (Suzgun et al., 2025b; Tang et al., 2025a; Yan et al., 2023) integrate heterogeneous memory forms including evolving cheatsheets, self-updating scientific libraries, and role-playing histories to achieve adaptability in specialized environments. However, these systems primarily optimize for *inter*-episode transfer which consolidates past success and failures to aid future attempts. This lacks the *intra*-episode mechanisms to recover if retrieved priors fail mid-exploration. ADAMEM bridges this gap by replacing static retrieval with dynamic, context-aware strategy synthesis.

## 3. Methodology

We propose **ADAMEM**, a framework that enables *test-time adaptation* for language agents. As illustrated in Figure 2, previous mechanisms like Synapse (Zheng et al., 2024a) and ReasoningBank (Ouyang et al., 2025) employ *static initialization*, where the agent retrieves a trajectory or strategy only at the episode start ($S_0$). This restricts the agent to fixed guidance throughout the task, often leading to failure when the initial plan becomes obsolete. In contrast, ADAMEM adapts the agent behaviors by dynamically updating its context with an up-to-date strategy as the short-term memory. In this section, we first describe the inference-time mechanism for generating adaptive strategy memory (§3.1), followed by our STEP-MFT technique that trains the agent to synthesize

this guidance effectively (§3.2).

### 3.1. ADAMEM: Test-Time Adaptive Memory

The core of ADAMEM is a non-parametric adaptation mechanism. As illustrated in Figure 1, the agent maintains a static pool of raw experiences ($\mathcal{M}$) but generates dynamic, state-specific strategies ($z_t$) that steer the agent decision-making without modifying parameters.

**Long-Term Trajectory Memory ($\mathcal{M}$).** We first construct a scalable long-term memory offline, denoted as $\mathcal{M}$. We define a trajectory as a sequence $\tau = \{(s_1, a_1, r_1), \ldots, (s_T, a_T, r_T)\}$, where $s_t$, $a_t$, and $r_t$ denote the state, action, and reward at step $t$, respectively. In practice, we include the full interaction history in the agent prompt to represent the state $s_t$, ensuring the model has access to all prior context for reliable decision-making. Given the sparse reward structures of common LLM agent applications where success is indicated only at the end of an episode, we populate $\mathcal{M}$ exclusively using trajectories that achieve a final success ($r_T = 1$). To enable dense retrieval, we index $\mathcal{M}$ by state embeddings. For every step $t$ in a successful trajectory, we store a key-value pair: $k_i = \phi(s_t)$, $v_i = (s_t, a_t, \tau_{t+1:T})$ where $\phi(\cdot)$ is a pretrained embedding model. The value $v_i$ contains the current state $s_t$, the immediate action $a_t$, and the remaining trajectory $\tau_{t+1:T}$ that leads to final success. This structure

constitutes the basis for adaptation, furnishing the agent with verified demonstrations of successful decision-making in similar historical states.

**Short-Term Strategy Memory ($z_t$).** At test time, the agent does not use $\mathcal{M}$ directly. Instead, it synthesizes a *Short-Term Strategy Memory*, denoted as $z_t$. This is a concise natural language strategy generated on-the-fly that serves as concrete guidance for the selection of the next action. Crucially, this differs from prior work like ReasoningBank (Ouyang et al., 2025), which relies on strategies that are pre-generated offline and based solely on past experiences. In contrast, our $z_t$ is generated *online* and is explicitly conditioned on the current state $s_t$, ensuring that the guidance is tailored to the specific dynamics of the test-time environment.

This synthesis process creates a natural control point for optimization by managing the lifetime of the strategy in the agent short-term memory. We introduce two inference modes of varying *adaptation effort* (Figure 2), allowing practitioners to balance the precision of fresh guidance against the token cost of regeneration. The full inference procedures for both modes are illustrated in Algorithm 1 of Appendix A.

**I. ADAMEM-HIGH (High Adaptation Effort).** This variant prioritizes adaptability by ensuring the strategy is always strictly conditioned on the immediate state. At step $t$, the agent outputs a tentative action $a'_t$ and a binary decision of memory retrieval $d_{\text{mem}} \in \{\text{yes}, \text{no}\}$ based on state $s_t$. If $d_{\text{mem}} = \text{no}$, the agent proceeds with the tentative action $a'_t$ without strategy conditioning. If $d_{\text{mem}} = \text{yes}$, the agent triggers memory retrieval to form the context $\mathcal{E}_{\text{ret}}$, synthesizes a fresh strategy $z_t \sim \pi_\theta(z|s_t, \mathcal{E}_{\text{ret}})$, and generates a refined action $a_t \sim \pi_\theta(a|s_t, z_t)$. Crucially, in this mode, $z_t$ is a transient strategy used solely for the current step and is immediately discarded from the context after usage. Consequently, if the agent requires guidance at the next step, it must perform the computationally expensive strategy synthesis process. While this incurs higher token costs, it maximizes robustness by preventing the agent from relying on potentially stale guidance.

**II. ADAMEM-LOW (Low Adaptation Effort).** To improve token efficiency, ADAMEM-LOW treats $z_t$ as a persistent state variable that guides multiple steps. Unlike ADAMEM-HIGH, which must regenerate guidance from scratch, ADAMEM-LOW maintains a currently active strategy $z_{\text{curr}}$ in its context. At step $t$, the agent predicts both a tentative action and a refresh decision $d_{\text{refresh}}$: $(a'_t, d_{\text{refresh}}) \leftarrow \pi_\theta(s_t, z_{\text{curr}})$. If $d_{\text{refresh}} = \text{no}$, the agent simply executes the tentative action $a'_t$, continuing to rely on the existing $z_{\text{curr}}$ if needed. Conversely, if $d_{\text{refresh}} = \text{yes}$, the agent retrieves relevant experiences $\mathcal{E}_{\text{ret}}$ and synthesizes a fresh strategy $z_{\text{new}}$. This strategy guides the immediate action $a_t \sim \pi_\theta(a|s_t, z_{\text{new}})$ and updates the persistent mem-

ory ($z_{\text{curr}} \leftarrow z_{\text{new}}$). By persisting the strategy, the agent can benefit from high-level guidance across multiple steps, effectively reducing the adaptation cost until a significant distribution shift requires an update.

## 3.2. STEP-MFT: Learning to Curate Strategy Memory

Although ADAMEM can operate via zero-shot prompting, untrained models often produce suboptimal strategies that fail to capture the nuances of dynamic environments. Therefore, we aim to train the policy to curate high-quality strategies that genuinely improve decision-making. This requires a training signal that distinguishes effective guidance from generic text. Standard outcome-based filtering (Uesato et al., 2022) is not reliable because it assumes all step-wise strategies on a successful trajectory are correct. This naive attribution credits irrelevant or misleading strategies that do not actively contribute to the success.

Formally, we seek to maximize the *Strategy Advantage*, denoted as $A(s, z)$, which quantifies the utility of a generated strategy $z$ given state $s$. Let $V^\pi(s)$ denote the value function (expected probability of success) starting from state $s$ under policy $\pi$. Treating the strategy $z$ as a high-level action, we define its advantage as the expected improvement in success rate over the baseline:

$$A(s, z) = \underbrace{V^{\pi_{\text{mem}}}(s)}_{\text{Value with Strategy } z} - \underbrace{V^{\pi_{\text{base}}}(s)}_{\text{Baseline Value}} \quad (1)$$

where $\pi_{\text{mem}}(\cdot) = \pi_\theta(\cdot|s, z)$ and $\pi_{\text{base}}(\cdot) = \pi_\theta(\cdot|s)$. Maximizing $A(s, z)$ isolates strategies that strictly increase the likelihood of success compared to the memory-free baseline.

However, directly estimating $V$ is computationally prohibitive in multi-turn scenarios, as it requires Monte Carlo rollouts (Wang et al., 2024b; Feng et al., 2025; Lee et al., 2025) or a specialized process reward model (Choudhury, 2025; Xi et al., 2025; Chae et al., 2025). To address this, we introduce **STEP-MFT**, which reinforces the generation of high-utility strategies that act as *necessary* conditions for correct decision-making. Concretely, STEP-MFT uses a computation-free proxy for $A(s, z)$ to achieve fine-grained credit assignment for strategy quality without auxiliary value models or environmental rollouts.

### 3.2.1. DATA COLLECTION

We perform inference on the training set using the ADAMEM-HIGH mechanism to collect a dataset of decision tuples $\mathcal{D}$. We select ADAMEM-HIGH over ADAMEM-LOW to maximize sample density: while ADAMEM-LOW caches persistent strategies to save tokens, yielding only sparse updates, ADAMEM-HIGH evaluates transient strategies at every step. This allows us to harvest a rich, dense set of diverse $(s_t, z_t)$ training pairs from each trajectory. The collected dataset consists of tuples

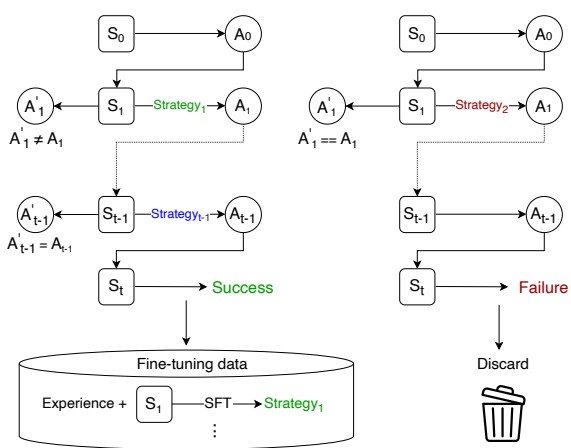

*Figure 3.* Overview of the **STEP-MFT** framework. We employ dual-filter rejection sampling to curate high-utility strategies for supervised fine-tuning (SFT). The process retains only *successful* trajectories where the strategy actually changes the proposed action ($A_t \neq A_t'$, green), while discarding redundant instances where the memory-free baseline yields the same action ($A_t = A_t'$, blue).

$\mathcal{D} = \{(s_t, \mathcal{E}_{\text{ret}}, z_t, a_t, a_t', r)\}$, where $s_t$ is the current state, $\mathcal{E}_{\text{ret}}$ is the retrieved context from Long-Term Memory $\mathcal{M}$, and $z_t$ is the generated strategy. We also record $a_t'$, the baseline action sampled from the policy $\pi_\theta(\cdot|s_t)$ without memory conditioning, and $a_t$, the action sampled from $\pi_\theta(\cdot|s_t, z_t)$ with memory conditioning. Finally, $r$ denotes the final trajectory reward. Crucially, $a_t'$ is a hypothetical action sampled for comparison purposes only; it is not executed in the environment.

### 3.2.2. PROCESS-LEVEL FILTERING

Our objective is to curate strategies that yield a strictly positive advantage over the baseline policy. In many steps of a successful trajectory, the baseline agent ($a_t'$) selects the correct action even without memory. Training on these redundant samples encourages reasoning that does not influence actions. To ensure the strategy actively drives decision-making, we filter for instances that alter the actions.

Specifically, we apply a dual-filter: (1) **Outcome Success:** the trajectory must be successful ($r = 1$) and (2) **Action Change**: the strategy must alter the agent decision on the next action ($a_t \neq a_t'$, see Figure 3). In practice, (2) is simply based on an exact lexical match between two action strings, which induces negligible overhead. By design, this filter prioritizes **precision over recall**: it may discard instances where the baseline action would also have succeeded, but ensures the curated set contains only strategies that actively drove successful decision-making. We formally justify this

condition as a prerequisite for positive advantage below.

**Proposition 3.1** (Necessity of Action Change). *Let $Q(s, a)$ be the action-value function representing the probability of task success. Assuming deterministic greedy decoding and that actions from successful trajectories are sufficiently effective, operationalized via sparse outcome rewards as a proxy for step-level correctness labels, for a strategy $z$ to have a strictly positive Advantage ($A(s, z) > 0$), it is necessary that the memory-conditioned action differs from the baseline action ($a_t \neq a_t'$).*

*Proof.* Under deterministic greedy decoding, the value of a state is entirely determined by the immediate action chosen by the policy: $V^\pi(s) = Q(s, \pi(s))$. Therefore, we can rewrite the Strategy Advantage from Equation 1 as:

$$A(s, z) = Q(s, a_t) - Q(s, a_t') \qquad (2)$$

Since our training data consists exclusively of *successful trajectories*, the action $a_t$ is empirically verified to lead to success. Rather than assuming strict optimality, we treat $a_t$ as sufficiently effective by using sparse outcome rewards as a proxy for process rewards, such that $Q(s, a_t) \geq Q(s, a_t')$. Consequently, the advantage is non-negative ($A(s, z) \geq 0$) by default.

To achieve a *strictly positive* advantage ($A(s, z) > 0$), we require $Q(s, a_t) > Q(s, a_t')$. We consider the counterfactual where the actions are identical ($a_t = a_t'$). In this case, the difference in action-value is strictly zero:

$$a_t = a_t' \implies Q(s, a_t) - Q(s, a_t') = 0 \implies A(s, z) = 0 \qquad (3)$$

Thus, a strictly positive advantage is impossible if the actions are unchanged. Therefore, $a_t \neq a_t'$ is a necessary condition for $A(s, z) > 0$. $\qquad \square$

### 3.2.3. TRAINING OBJECTIVE

The filtered dataset $\mathcal{D}^*$ consists of pairs $((s, \mathcal{E}_{\text{ret}}), z^*)$ where $z^*$ represents a "silver" strategy that passes the process-level filtering. We instantiate this learning process via rejection sampling fine-tuning (RFT; Yuan et al., 2023), optimizing the policy using standard cross-entropy loss used in supervised fine-tuning (SFT) to maximize the likelihood of these effective strategies:

$$\mathcal{L}_{\text{SFT}} = -\mathbb{E}_{(s, \mathcal{E}, z^*) \sim \mathcal{D}^*}[\log \pi_\theta(z^*|s, \mathcal{E})] \qquad (4)$$

While our process-level reward could in principle be optimized using online reinforcement learning (RL) algorithms such as GRPO (Shao et al., 2024), we opt for RFT to efficiently inject the process-level signals without the computational overhead and instability associated with online training. We leave the investigation of full memory-augmented RL (Jiang et al., 2025a; Liu et al., 2026) to future work.

To minimize deployment complexity, we employ this single fine-tuned model for both strategy and action generation (Stages 3 and 4 in Figure 1), eliminating the need to host separate specialized models during inference. We refer to this unified, fine-tuned agent as ADAMEM-MFT.

## 4. Experiments

### 4.1. Experimental Setup

**Agent Benchmarks.** We evaluate ADAMEM on three benchmarks spanning embodied navigation, web shopping, and agentic search. **ALFWorld** (Shridhar et al., 2021) features embodied household tasks across six categories, such as *Pick & Place* and *Heat & Place*. Agents must navigate and interact with objects to fulfill high-level goals. We assess generalization using *seen* (140 instances) and *unseen* (134 instances) splits, the latter containing entirely novel room layouts. Performance is measured by the **Success Rate** (%). **WebShop** (Yao et al., 2022) simulates an e-commerce platform with 1.1 million products. The agent navigates HTML interfaces to ground natural language instructions into purchase actions. We use the dense metric **Task Score** (0 to 100) to measure how closely the selected product's attributes and price match the user's requirements. **HotpotQA** (Yang et al., 2018) is a multi-hop question answering benchmark that requires the agent to reason over multiple documents to answer complex factual questions. To extend the task horizon of agentic search, we adopt the *cross-episode* evaluation setup (Xiao et al., 2026; Jiang et al., 2025b), where an agent iteratively refines its answer across multiple episodes. At the end of each episode, a self-reflection step is always triggered, prompting the agent to reconsider its current answer and search for additional evidence in the next episode. We evaluate on 500 test questions with a per-question budget of at most 15 reasoning steps or 3 episodes, whichever is reached first. Performance is measured by **Success Rate** (%), which is the percentage of tasks where the agent answer exactly matches the gold answer. All benchmarks are evaluated in a text-only setting.

**Backbone Models.** We adopt **Qwen3-4B-Instruct-2507** (Yang et al., 2025) for ALFWorld and HotpotQA, and an RL-trained **Qwen2.5-7B-Instruct** (Yang et al., 2024a) for WebShop.[2] We select the RL-trained 7B model for Web-Shop due to the insufficient success rate of 4B base model on this task, which hinders the collection of successful trajectories required to populate the long-term memory. Our primary evaluation employs an **on-policy** setup where the same backbone constructs memory, synthesizes strategies, and generates actions. This strictly tests for *self-improvement* rather than distillation. To assess *cross-model generalizabil-*

*ity*, we introduce an **off-policy** setup using **Gemma-27b-it**. Representing a distinct model family and size, Gemma synthesizes strategies and generates actions using frozen memory banks created by the Qwen models, thereby testing the transferability of long-term memory across different models. Qwen3-Embedding-4B (Zhang et al., 2025b) provides embedding for all retrieval operations.

**Memory Baselines.** We compare ADAMEM against several representative agent memory baselines: (1) **No Memory**, the base LLM agent operating without any memory module; (2) **Synapse** (Zheng et al., 2024a), a representative method that retrieves full past trajectories to serve as in-context exemplars; and (3) **ReasoningBank** (Ouyang et al., 2025), which distills raw trajectories into high-level strategies *offline*.[3] ReasoningBank can be viewed as a static special case of ADAMEM, distinguished by two key rigidities: strategies are *pre-computed offline* rather than synthesized online conditioned on the current state, and retrieval is restricted to *episode initiation* rather than adapting dynamically to intermediate steps. We evaluate in the *offline* setting (Wang et al., 2025), where the long-term memory is pre-populated with successful trajectories or strategies on training data.

**Implementation Details.** We implement all agents using the **ReAct** (Yao et al., 2023) framework, as its interleaved reasoning and action traces significantly outperform zero-shot direct action generation in long-horizon tasks. For the Long-Term Trajectory Memory, we populate the index with valid trajectories collected from the training set, resulting in 1,596 correct trajectories for ALFWorld, 7,150 for WebShop and 965 for HotpotQA. To strictly control for information access, all compared memories are constructed based on this identical pool of raw successful trajectories. Unless otherwise specified, all tables and figures use top-1 ($k = 1$) retrieval by default. We study the effect of varying $k$ in Figure 4. To train the STEP-MFT agent, we curate a dataset of 13,067 filtered strategies for ALFWorld, and 12,492 for WebShop. Detailed training and inference configurations are provided in Appendix A. We include prompts for our ADAMEM and baseline agents in Appendix B.

### 4.2. Main Results

**ADAMEM outperforms static memory baselines across diverse agentic tasks.** Table 1 presents the evaluation results under the training-free setup. ADAMEM-LOW consistently outperforms baselines across all settings, with the most pronounced gains observed in generalization scenarios. On the challenging ALFWorld unseen split, ADAMEM-LOW achieves a substantial +11.4 point absolute improve-

---

[2]We use the checkpoint *langfeng01/GiGPO-Qwen2.5-7B-Instruct-WebShop* released by Feng et al. (2025).

[3]We exclude workflow-based memories like AWM (Wang et al., 2025), because the selected benchmarks operate within fixed workflows and the primary challenge is *adapting* to dynamic situations.

*Table 1.* Performance comparison under *training-free* setups. We report the Success Rate (%) for ALFWorld across seen and unseen splits and the average Task Score for WebShop. Results are reported as mean ± standard deviation over 3 independent runs. **Bold** indicates the highest score, and underline denotes the second highest. ADAMEM under its low adaptation effort outperforms static memory baselines, demonstrating strong generalization on unseen tasks and across different model families.

| Memory Mechanisms | Long Term | Short Term | ALFWorld Seen | ALFWorld Unseen | WebShop |
|---|---|---|---|---|---|
| *On-Policy* **Long-Term Memory** | | | | | |
| 🦅 *Qwen3-4B-Instruct (ALFWorld)* | | | | | |
| 🦅 *Qwen2.5-7B-Instruct-RL (WebShop)* | | | | | |
| No Memory | - | - | $45.2_{\pm1.8}$ | $46.8_{\pm2.5}$ | $71.4_{\pm1.4}$ |
| ReasoningBank | Strat. | Strat. | $49.3_{\pm0.7}$ | $51.2_{\pm0.9}$ | $68.6_{\pm2.0}$ |
| Synapse | Traj. | Traj. | $52.1_{\pm1.9}$ | $52.2_{\pm0.7}$ | $65.4_{\pm0.9}$ |
| ADAMEM | Traj. | Strat. | **$54.0_{\pm2.9}$** | **$58.2_{\pm3.9}$** | **$74.2_{\pm0.3}$** |
| *Off-Policy* **Long-Term Memory** | | | | | |
| 🔴 *Gemma-3-27b-it* | | | | | |
| No Memory | - | - | $36.7_{\pm0.8}$ | $37.6_{\pm1.9}$ | $18.2_{\pm0.1}$ |
| ReasoningBank | Strat. | Strat. | $45.2_{\pm2.2}$ | $44.8_{\pm1.3}$ | $18.6_{\pm0.5}$ |
| Synapse | Traj. | Traj. | $35.0_{\pm2.1}$ | $30.6_{\pm0.7}$ | $22.6_{\pm0.5}$ |
| ADAMEM | Traj. | Strat. | **$47.4_{\pm2.2}$** | **$49.5_{\pm2.6}$** | **$24.7_{\pm1.2}$** |

*Table 2.* Results on HotpotQA under the cross-episode agentic search setup with *Qwen3-4B-Instruct-2507*. We report Success Rate (%) as mean ± std over 3 independent runs. ADAMEM nder its low adaptation effort outpe consistently outperforms static memory baselines in this iterative search setting.

| Memory Mechanisms | Long Term | Short Term | HotpotQA SR (%) |
|---|---|---|---|
| No Memory | - | - | $39.7_{\pm1.9}$ |
| Synapse | Traj. | Traj. | $40.4_{\pm0.2}$ |
| ReasoningBank | Strat. | Strat. | $40.5_{\pm0.8}$ |
| ADAMEM | Traj. | Strat. | **$41.1_{\pm0.5}$** |

ment over the "No Memory" baseline and surpasses the strongest static memory method (Synapse) by +6.0 points without any training. This significant margin validates our method's ability to retrieve and adapt high-level strategies to novel environments. ADAMEM also achieves the best performance on HotpotQA (Table 2), an iterative multi-hop search task evaluated under a cross-episode setup. Across all three tasks, static methods with fixed retrieval timing consistently underperform ADAMEM, confirming that dynamic memory adaptation is effective regardless of task structure.

**Dynamic retrieval corrects negative transfer on WebShop.** A critical insight from Table 1 is the failure of static memory on WebShop, where baselines suffer from negative transfer. Specifically, Synapse and ReasoningBank underperform the memory-free ReAct agent by 6.0 and 2.8 points, respectively. We attribute this deficit to their rigid retrieval

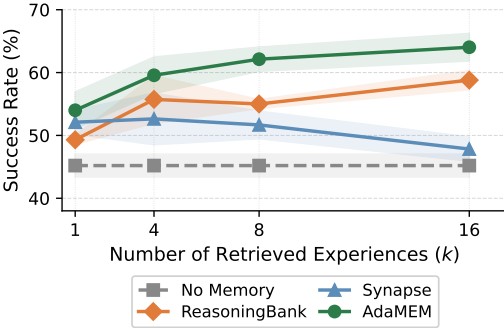

*Figure 4.* Scalability with retrieval budget $k$ on ALFWorld seen split (Qwen3-4B-Instruct, on-policy). Shaded bands show ±std over 3 runs. ADAMEM scales monotonically with $k$, while Synapse degrades as injecting additional raw trajectories into the context leads to context overflow and increased interference from irrelevant content.

timing at episode initiation ($t = 0$), where the uninformative homepage state leads to the retrieval of noisy, irrelevant trajectories. ADAMEM reverses this trend: by dynamically triggering updates mid-trajectory, it effectively leverages context-rich states (e.g., search results) to achieve a +2.8 point gain over the memory-free baseline.

**Hybrid architecture enables effective memory sharing despite off-policy distribution shifts.** We demonstrate that our framework accommodates diverse backend models and allows them to share a unified long-term memory. In our *off-policy* evaluation (Table 1), Gemma-3-27b-it performs inference using trajectory memory banks originally constructed by Qwen models. Despite the distribution shift between the behaviors of the generator and the policy, ADAMEM-LOW delivers robust improvements. It boosts WebShop performance by +6.5 points over the "No Memory" baseline and outperforms Synapse by +2.1 points. This confirms the flexibility of our hybrid memory architecture: by decoupling storage (trajectories) from adaptation (strategy synthesis), ADAMEM ensures that the strategy is synthesized *by the current policy*, tailored to its own capabilities. This avoids the limitation of baselines like ReasoningBank, where strategies are pre-generated offline by the source model, forcing the inference agent to follow reasoning traces that may be incompatible with its own.

**ADAMEM scales consistently with more retrieved experiences.** Figure 4 reports ALFWorld (seen) success rate as the retrieval budget $k$ increases from 1 to 16. ADAMEM benefits monotonically from additional retrieved experiences, improving from 54.0% at $k$=1 to 64.0% at $k$=16 (+10 points). ReasoningBank also improves with $k$ but plateaus and gains less (+9.5 points), while Synapse *degrades* at larger $k$ (52.1% → 47.8%). We attribute this to

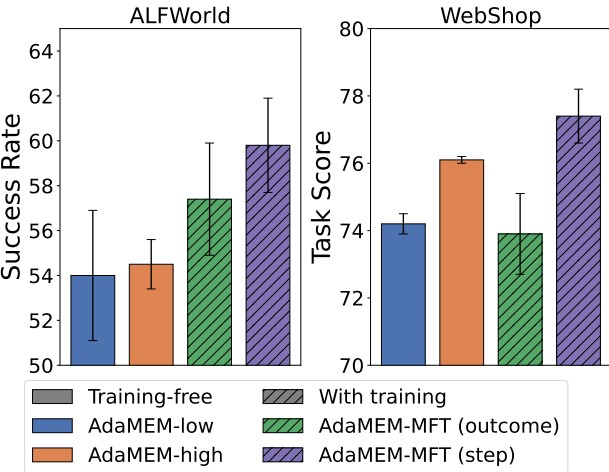

*Figure 5.* Impact of Memory Fine-Tuning (MFT). Following the *on-policy long-term memory* setup in Table 1, we use Qwen3-4B-Instruct for ALFWorld seen split and Qwen2.5-7B-Instruct-RL for WebShop. We compare training-free baselines against MFT variants trained with *Outcome*-only and *Outcome + Action Change* (step) dual-filtering. STEP-MFT consistently outperforms outcome-based MFT and training-free baselines.

*Figure 6.* Effectiveness vs. Efficiency trade-off. Following the *on-policy long-term memory* setup in Table 1, we use Qwen3-4B-Instruct for ALFWorld seen split and Qwen2.5-7B-Instruct-RL for WebShop. We plot task performance (y-axis) against the average inference cost in tokens per step (x-axis). ADAMEM establishes a superior Pareto frontier compared to static memory baselines.

the verbosity of raw trajectory retrieval: as $k$ increases, concatenating multiple full trajectories substantially expands the context, overwhelming the agent with redundant and irrelevant content. In contrast, ADAMEM compresses retrieved experiences into concise strategies, avoiding context overflow and enabling consistent gains with scale. This confirms that strategy abstraction is the key ingredient for effectively leveraging a larger memory bank.

**Step-wise filtering is critical for learning effective strategy memory.** While training-free variants perform well, Figure 5 demonstrates that training for memory generation enables better test-time adaptation. We compare two filtering strategies for our STEP-MFT framework: filtering by **Outcome Success** (keeping strategies from all successful trajectories) versus additionally filtering by **Action Change** (keeping only strategies that causally altered the agent's action). To ensure fair comparisons, we keep the fine-tuning examples for both methods the same. The results reveal the dangers of naive outcome-based supervision: on Web-Shop, filtering by Outcome Success actually degrades performance compared to the training-free ADAMEM-HIGH baseline (73.9 vs. 76.1). In contrast, the STEP-MFT consistently drives monotonic improvements, achieving a clear +1.3 point gain over ADAMEM-HIGH. The advantage is even more pronounced on ALFWorld, where step-wise filtering yields a substantial +5.3 point improvement over the training-free ADAMEM-HIGH baseline (59.8 vs. 54.5), confirming that training with fine-grained process-level signals

is essential. The higher token consumption of ADAMEM-MFT relative to ADAMEM-HIGH is a direct consequence of this training: by learning exclusively on instances where dynamic synthesis was decisive, ADAMEM-MFT develops a stronger propensity to trigger memory generation, increasing the strategy update frequency from 34% to 36% on ALFWorld and from 63% to 74% on WebShop (see Appendix D).

**ADAMEM effectively leverages additional test-time compute.** Figure 6 illustrates the relationship between task performance and inference cost. ADAMEM demonstrates a robust positive scaling trend with respect to test-time compute: as we increase the inference compute (by increasing memory update frequency from ADAMEM-LOW to ADAMEM-HIGH and ADAMEM-MFT), performance improves monotonically. This contrasts sharply with the negative scaling of baselines like Synapse on WebShop, which consume significantly more tokens than ReAct but yield *lower* task scores. By effectively converting additional test-time compute into improved decision-making, our framework avoids the stagnation characteristic of rigid retrieval heuristics, allowing for continuous improvement proportional to the available computational budget. This efficiency extends to latency as well. In Table 4 and Appendix D, we demonstrate that ADAMEM reduces the inference latency by 16% compared to Synapse, as synthesizing concise strategies reduces the need for processing lengthy raw trajectory logs.

*Table 3.* Ablation of memory components on ALFWorld seen split with Qwen3-4B-Instruct, using on-policy long-term memory. Results confirm that both long-term memory (retrieving past trajectory) and short-term memory (abstracting strategy) are essential to maximize success.

| Systems | Success Rate (↑) | # Inference Tokens Per Step (↓) |
|---|---|---|
| ADAMEM-MAX | $\mathbf{65.5}_{\pm 3.2}$ | 6.0K |
| w/o strategy memory | $59.3_{\pm 5.0}$ | 3.8K |
| w/o trajectory memory | $34.8_{\pm 2.7}$ | 3.9K |
| w/o both (no memory) | $45.2_{\pm 1.8}$ | 2.1K |

### 4.3. Ablation Study

To understand the specific contribution of each component in the ADAMEM framework, we conduct an ablation study on ALFWorld (Table 3). To eliminate the variance introduced by the agent's autonomous decision on *when* to refresh, we enforce a strategy refresh at every step for this experiment, resulting in a special variant ADAMEM-MAX. This ensures that at every time step $t$, the agent queries the memory, synthesizes a strategy, and decides the next action.

**Necessity of Abstraction with Short-Term Memory.** To investigate whether the explicit *abstraction* of a strategy is necessary, or if the agent can simply imitate raw retrieved examples, we evaluate a variant *w/o short-term strategy memory*. In this setting, the agent retrieves raw experiences based on the current state at every step but bypasses strategy synthesis, feeding the retrieved logs directly to the policy. While this variant reduces inference cost (6.0K → 3.8K tokens/step) by eliminating the intermediate generation step of synthesizing $z_t$, the performance drops from 65.5% to 59.3%. This confirms that simply retrieving raw trajectories is insufficient; the agent requires the Short-Term Memory to *abstract* diverse experiences into actionable, state-specific guidance, a benefit that outweighs the additional generation cost.

**Necessity of Grounding with Long-Term Memory.** To investigate if the agent can effectively strategize using only its internal knowledge without external *grounding*, we evaluate a variant *w/o long-term trajectory memory*. Here, the agent attempts to synthesize a strategy solely based on the current task state. This variant sees a reduction in token usage (6.0K → 3.9K tokens/step) because the agent's context window is not burdened by lengthy retrieved trajectories ($\mathcal{E}_{\text{ret}}$). However, this efficiency comes at a severe cost: performance drops to 34.8%, significantly worse than even the memory-free ReAct baseline (45.2%). Manual observation in Appendix C.1 reveals that without the *grounding* provided by retrieved successful trajectories, the agent's self-generated strategies often degenerate into hallucinations based on misunderstandings of environmental feedback.

### 4.4. Case Study

Figure 7 (Appendix C.2) illustrates ADAMEM overcoming distribution shifts in ALFWorld. When an initial strategy incorrectly directs the agent to cabinets (while soapbars are on the countertop), static agents often trap themselves in empty containers. In contrast, ADAMEM detects the lack of progress, explicitly reasons that the *"strategy is outdated,"* and triggers a refresh to correctly redirect search. This confirms that step-wise updates enable self-correction against stale priors. However, Figure 8 (Appendix C.3) reveals a limitation where the agent fails to trigger a necessary refresh, incorrectly rationalizing the failure as merely incomplete exploration rather than a flawed plan.

## 5. Conclusion

We propose ADAMEM, a hybrid memory framework that addresses the rigidity of static retrieval in agent memory. By decoupling long-term trajectory storage from short-term strategy synthesis, ADAMEM enables agents to adapt their reasoning to evolving task states without online parameter updates. We further introduce STEP-MFT, a rejection-sampling fine-tuning technique that trains agents to generate high-utility strategies based on fine-grained, process-level impacts on decision-making. Empirically, our approach establishes a superior Pareto frontier between inference efficiency and task performance, with consistent gains across three agentic benchmarks compared to static memory baselines. This work demonstrates that dynamic memory curation serves as a critical scaling dimension for agentic reasoning in open-ended environments.

## Acknowledgments

This work was supported by Cisco Research, and computational resources and services provided by Advanced Research Computing (ARC), a division of Information and Technology Services (ITS) at the University of Michigan, Ann Arbor. This work used the Bridges-2 system at the Pittsburgh Supercomputing Center (PSC). We also thank the ICML reviewers and the members of the LAUNCH group at the University of Michigan for their constructive feedback.

## Impact Statement

This paper presents ADAMEM, a framework for test-time adaptive memory in language agents. By enabling agents to continuously refine their behavior from accumulated experience without parameter updates, this work contributes to more capable and efficient AI systems for long-horizon tasks such as embodied navigation, web interaction, and information retrieval.

The primary societal benefits lie in reducing the cost of deploying adaptive agents: practitioners can improve agent performance post-deployment without expensive retraining, lowering barriers to building capable AI assistants. The framework's non-parametric nature also makes it more transparent, as the long-term memory consists of interpretable natural language trajectories and strategies.

At the same time, agents that learn from and act upon accumulated experience raise important considerations. Memory banks populated from user interactions may inadvertently retain sensitive information, requiring careful data governance. Additionally, more capable autonomous agents operating in open-ended environments (e.g., web navigation) warrant robust oversight mechanisms to prevent unintended actions. We encourage practitioners deploying systems built on this framework to consider appropriate access controls, memory auditing, and human-in-the-loop safeguards proportional to the deployment context.

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

# A. Implementation Details

**Inference Configurations.** Algorithm 1 formally defines the inference loop for both ADAMEM-HIGH and ADAMEM-LOW modes. We build a semantic index with `hnswlib` (Malkov & Yashunin, 2016), using the full `Qwen3-Embedding-4B` output as state embeddings (dimension 2560). The HNSW index is built in cosine space over L2-normalized embeddings with $M=32$, `ef_construction=200`, and `ef_search=100`. During inference, we use a decoding temperature of 0.7 and maximum generation tokens of 2048 for all models. We use the ALFWorld, WebShop, and HotpotQA environments provided by `verl-agent` (Feng et al., 2025).

**Training Configurations.** We implement STEP-MFT using LLaMA-Factory (Zheng et al., 2024b), fine-tuning the backbone via LoRA (Hu et al., 2022) with a rank of 32. Optimization employs a global batch size of 64, a learning rate of with a cosine scheduler (0.1 warmup ratio), and proceeds for 1 epoch. All experiments utilize NVIDIA A40/L40S GPUs (48GB).

# B. Agent Prompts

## B.1. Baseline Agents

---
**Prompt for the ReAct Agent (No Memory)**

You are an expert agent operating in the ALFRED Embodied / WebShop e-commerce Environment. Your task is to: {task_description}
Prior to this step, you have already taken {step_count} step(s). Below are the most recent {history_length} observations and the corresponding actions you took: {action_history}
You are now at step {current_step} and your current observation is: {current_observation}
Your admissible actions of the current situation are: [{admissible_actions}].
Now it's your turn to take an action.
You should first reason step-by-step about the current situation. This reasoning process MUST be enclosed within <think></think> tags.
Once you've finished your reasoning, you should choose an admissible action for current step and present it within <action></action> tags.

---
**Prompt for the ReasoningBank Agent**

{react_prompt}
Below are some strategy items that are accumulated from past interactions from the environment that may be helpful to solve the task. You can use it when you feel it's relevant. In each step, please first explicitly discuss if you want to use each strategy item or not, and then take action.
{retrieved_strategies}

---

**Algorithm 1** Inference Process for ADAMEM

1: **Input:** Initial observation $o_0$, Long-term Memory $\mathcal{M}$, Policy $\pi_\theta$, Mode $\in$ {ADAMEM-HIGH, ADAMEM-LOW}
2: **Initialize:** History $h_0 \leftarrow \emptyset$, Persistent Strategy $z_{\text{curr}} \leftarrow \emptyset$ (for ADAMEM-LOW)
3: **for** $t = 0, 1, \ldots, T$ **do**
4:     $s_t \leftarrow (h_t, o_t)$
5:     **if** Mode == ADAMEM-HIGH **then**
6:         Sample tentative action and decision: $(a', d_{\text{mem}}) \sim \pi_\theta(s_t)$
7:         **if** $d_{\text{mem}} = $ Yes **then**
8:             Retrieve context: $\mathcal{E}_{\text{ret}} \leftarrow \text{Retrieve}(s_t, \mathcal{M})$
9:             Consolidate strategy: $z_t \sim \pi_\theta(z \mid s_t, \mathcal{E}_{\text{ret}})$
10:             Generate adapted action: $a_t \sim \pi_\theta(a \mid s_t, z_t)$
11:         **else**
12:             $a_t \leftarrow a'$
13:         **end if**
14:     **else if** Mode == ADAMEM-LOW **then**
15:         Sample action and refresh decision: $(a_{\text{init}}, d_{\text{refresh}}) \sim \pi_\theta(s_t, z_{\text{curr}})$
16:         **if** $d_{\text{refresh}} = $ Yes **then**
17:             Retrieve context: $\mathcal{E}_{\text{ret}} \leftarrow \text{Retrieve}(s_t, \mathcal{M})$
18:             Update persistent strategy: $z_{\text{curr}} \sim \pi_\theta(z \mid s_t, \mathcal{E}_{\text{ret}})$
19:             Re-generate action: $a_t \sim \pi_\theta(a \mid s_t, z_{\text{curr}})$
20:         **else**
21:             $a_t \leftarrow a_{\text{init}}$
22:         **end if**
23:     **end if**
24:     Execute $a_t$, receive observation $o_{t+1}$
25:     $h_{t+1} \leftarrow h_t \cup \{(a_t, o_{t+1})\}$
26: **end for**

---

**Prompt for the Synapse Agent**

{react_prompt}
Here are some task solving trajectories on similar tasks:
{retrieved_experiences}

---

## B.2. ADAMEM Agent

---
**Prompt for the ADAMEM-HIGH Agent: Memory Retrieval Decision**

{react_prompt}
MEMORY RETRIEVAL OPTION:
You have access to a memory system that stores past experiences from similar situations. Using memory can often clarify ambiguity and suggest effective strategies.
After presenting your reasoning and initial proposed action, consider whether retrieving related past experiences could improve your decision. Retrieval may return both successful and failed examples—successful ones illustrate proven

---

approaches, while failed ones highlight common pitfalls to avoid. Both are valuable for informed decision-making.
We recommend retrieving memory in ambiguous or uncertain situations, such as when:
- Multiple possible actions exist without a clear best choice.
- You are unsure about the next step (your reasoning includes words like "maybe," "uncertain," or "not sure").
- The task involves a sequence of actions where past examples could serve as a concrete template.
Your output format should be:
<think>...your reasoning...</think>
<action>...your chosen action...</action>
<memory_request_rationale>Briefly explain whether and why memory would help here (1-2 sentences)</memory_request_rationale>
<request_memory>yes or no</request_memory>

---

### Prompt for the ADAMEM-LOW Agent: Memory Refresh Decision

{react_prompt}
Below are some strategy items that are accumulated from past interactions from the environment that may be helpful to solve the task. You can use it when you feel it's relevant. In each step, please first explicitly discuss if you want to use each strategy item or not, and then take action.
{current_strategy}

___

### STRATEGY REFRESH OPTION
If you believe the current strategy is outdated or misaligned, you may request to synthesize a new strategy using your memory system.
**Implication of Refreshing:**
If you trigger a refresh ('yes'), your currently proposed action will **not** be executed. Instead, the system will update the strategy, and you will be given the opportunity to **make a new decision for this exact step** using the updated context.
**Recommended Criteria for Refreshing:**
* **Observation Mismatch:** The current state/observation contradicts the expectations set by your existing strategy.
* **Persistent Failure:** You have attempted similar actions repeatedly without making progress toward the goal.
* **Critical Ambiguity:** The current situation presents a novel obstacle or edge case where you lack the context to proceed safely.
**Instructions:**
1. Generate a **proposed action** based on the *current* strategy.
2. Decide if a refresh is strictly necessary based on the criteria above.
**Output format:**
<think>...reasoning...</think>
<action>...your proposed action...</action>
<refresh_decision>yes or no</refresh_decision>

---

### Prompt for the ADAMEM Agent: Strategy Memory Synthesis

You need to synthesize contextualized, non-generic strategies from retrieved past experiences. You are given K = {k} retrieved experiences, each containing the past state, past

action, remaining trajectory, and final outcome ("success" or "failure").
Goal: Derive a concise, actionable, and non-generic strategy tailored to the current state by learning from the retrieved experiences. This strategy will later guide the next action.
Constraints:
- Be specific but not overfitted — avoid instance-specific shortcuts unless clearly relevant to the current state.
- If the evidence from retrieved examples is weak or conflicting, briefly note that uncertainty within the strategy.
- Do not quote or copy from retrieved experiences; abstract their key insights.
Retrieved Experiences:
{retrieved_experiences}
First, provide your reasoning enclosed in <think>...</think>. Analyze how the current state resembles or differs from each past state, and explain why successful examples likely worked and why failed ones did not.
Then, within a single <strategy> block, list 1–3 concise, actionable, context-specific strategy bullets tailored to the current state. Do not produce multiple <strategy> tags. Do not generate the next action itself.
Output format:
<think>...your reasoning...</think>
<strategy>
- ...
- ...
- ...
</strategy>

---

### Prompt for the ADAMEM Agent: Strategy-Conditioned Action Generation

{react_prompt}
Below are some strategy items that are accumulated from past interactions from the environment that may be helpful to solve the task. You can use it when you feel it's relevant. In each step, please first explicitly discuss if you want to use each strategy item or not, and then take action.
{strategy}

## C. Case Study

### C.1. Strategy Hallucination

To investigate the performance degradation observed in the *w/o long-term memory* variant (Table 3), we present a qualitative analysis of a failure trajectory that illustrates how the absence of external grounding leads the agent to hallucinate inadmissible actions.

For instance, in a task to *"put a knife in sidetable,"* the correct action is to use the verb `move`. The agent had previously attempted to `move knife 1 to sidetable 1` multiple times but failed (receiving *"Nothing happens"*) because it had not yet navigated to the correct location. Now located at the sidetable, the agent reflects on this history without the benefit of external examples. It incorrectly infers that the verb `move` itself is invalid due to the repeated

failures. Consequently, it hallucinates that the correct verb must be `put` and generates a strategy explicitly instructing: *"- put knife 1 on sidetable 1."* Since `put` is not an admissible action key in this environment, this reasoning actively misleads the agent, whereas a simple ReAct agent might have successfully attempted `move` again.

### C.2. Success Case with Dynamic Adaptation

To demonstrate the efficacy of ADAMEM's adaptive mechanism, we analyze a specific instance from the ALFWorld benchmark shown in Figure 7 (Task: *"put two soapbar in garbagecan"*). This case illustrates how the agent recovers from both information misalignment and physical constraints through step-wise strategy updates.

**Phase 1: Initial Misalignment and Correction.** At the beginning of the episode, the agent retrieves an initial strategy based on the task description, which suggests systematically checking cabinets. The agent executes this plan, opening Cabinets 1 through 4. However, this initial strategy relies on a distributional assumption that soapbars are stored in cabinets. Upon finding all cabinets empty, a static memory agent (like Synapse or ReasoningBank) would likely hallucinate further cabinets or loop repetitively. In contrast, ADAMEM recognizes the lack of progress. The agent's thought process notes, *"the existing strategy is outdated because it assumes all soapbars are in cabinets,"* triggering a refresh decision ($d_{\text{refresh}} = $ yes).

**Phase 2: Strategy Shift.** The agent queries the Long-Term Trajectory Memory with the updated state (cabinets empty) and synthesizes a new strategy: *"Go to countertop 1... If not found, go to sinkbasin."* This directs the agent to the correct location immediately, where it successfully finds and picks up the first soapbar.

**Phase 3: Handling Environmental Constraints.** A critical adaptation occurs when the agent attempts to pick up the second soapbar immediately after the first. The environment renders the "take" action inadmissible (likely due to an inventory limit of one item). A standard agent often fails here, repeatedly trying to "take" the object despite the error. ADAMEM, however, detects that the action is not in the admissible list. It triggers a second strategy refresh, explicitly diagnosing the issue: *"The environment does not allow taking soapbar 2... a refresh is necessary."* The newly synthesized strategy corrects the workflow to a sequential execution: *"move soapbar 1 to [garbagecan]... return to countertop to retrieve the second soapbar."*

This trajectory demonstrates that ADAMEM does not merely follow a script; it actively monitors the validity of its strategy against environmental feedback, pivoting effectively when the current plan becomes obsolete or impossible.

*Table 4.* Comparison of inference latency on ALFWorld using Qwen3-4B-Instruct. Relative changes compared to the "No Memory" baseline are highlighted in color. ADAMEM achieves the best success rate with a moderate latency increase, effectively outperforming Synapse which suffers from a severe 50% latency penalty.

| | Memory Adaptation (seconds, ↓) | Total Time Per Step (seconds, ↓) | Success Rate (%, ↑) |
|---|---|---|---|
| No Memory | - | 28.5 | 45.2 |
| ReasoningBank | - | 29.8 ↑5% | 49.3 ↑9% |
| Synapse | - | 42.7 ↑50% | 52.1 ↑15% |
| ADAMEM | 5.7 | 36.8 ↑29% | 54.0 ↑19% |

### C.3. Failure Case with Strategy Inertia

To identify limitations in autonomous adaptation, we analyze a specific failure instance of ADAMEM-LOW on the ALFWorld task *"put a clean butterknife in countertop,"* as illustrated in Figure 8.

The episode begins with a reasonable strategy: systematically exploring cabinets and drawers. However, a critical failure mode emerges when this initial plan yields no results. After searching all primary containers (Cabinets 1–17, Drawers 1–7), the agent encounters a state of high uncertainty. Theoretically, this is the optimal moment to trigger a strategy refresh ($d_{\text{refresh}} = $ yes) to query the long-term memory for alternative object locations (e.g., searching exposed surfaces like tables or shelves).

Instead, the agent exhibits *strategy inertia*. As detailed in the thought trace, the agent rationalizes the lack of progress as incomplete exploration rather than a misalignment of the strategy itself, explicitly concluding: *"I do not believe a strategy refresh is necessary... failure to find the butterknife so far is due to incomplete exploration."* Consequently, it outputs $d_{\text{refresh}} = $ no and persists with the obsolete guidance, spiraling into a loop of searching low-probability locations (e.g., the fridge and sinkbasin).

This case demonstrates that while ADAMEM-LOW improves efficiency, it relies heavily on the agent's self-evaluation capabilities. When the model acts with overconfidence in a failing plan, it effectively locks itself out of the corrective guidance stored in the long-term memory, leading to episode failure via step exhaustion.

## D. Efficiency Analysis

We investigate the computational cost of test-time adaptation by measuring the average wall-clock time per step on ALFWorld. As shown in Table 4, ADAMEM achieves a superior trade-off between latency and performance compared to static memory baselines.

While the memory adaptation mechanism introduces an

*Table 5.* Average total tokens consumed per task on ALFWorld (Qwen3-4B-Instruct, on-policy). ADAMEM-LOW achieves the best token efficiency among memory-augmented methods.

| | Total Tokens Per Task (↓) | Success Rate (%, ↑) |
|---|---|---|
| No Memory | 77K | 45.2 |
| ReasoningBank | 81K ↑5% | 49.3 ↑9% |
| Synapse | 117K ↑52% | 52.1 ↑15% |
| ADAMEM-LOW | 90K ↑17% | 54.0 ↑19% |
| ADAMEM-HIGH | 150K ↑95% | 54.5 ↑21% |
| ADAMEM-MFT | 165K ↑114% | 65.5 ↑45% |

overhead of 5.7 seconds per step, the total inference time for ADAMEM (36.8s) remains significantly lower than that of Synapse (42.7s). This 16% reduction in latency compared to Synapse is primarily due to our strategy synthesis: whereas Synapse forces the agent to process long, token-heavy raw trajectories at every step, ADAMEM condenses this information into concise textual guidance.

Although ADAMEM increases total time by 29% relative to the memory-free baseline, this investment yields the highest return, boosting the success rate by 19% relative. In contrast, ReasoningBank offers lower latency but fails to achieve comparable success gains, while Synapse incurs a heavy 50% latency penalty for diminishing returns. These results confirm that ADAMEM effectively converts computational time into actionable reasoning, establishing a more efficient Pareto frontier for agentic memory.

**Token budget per task.** Table 5 reports the average total tokens consumed per task across ADAMEM variants. ADAMEM-LOW consumes 23% fewer tokens than Synapse (90K vs. 117K) while achieving a higher success rate, confirming that strategy synthesis is a more token-efficient way to leverage memory than raw trajectory injection. As adaptation effort increases from ADAMEM-LOW to ADAMEM-HIGH and ADAMEM-MFT, the token budget grows correspondingly, but so does task performance — reflecting a smooth and controllable compute-quality trade-off.

**Strategy update frequency.** Table 6 reports how often each ADAMEM variant triggers a strategy update per task. ADAMEM-LOW updates sparingly (avg. 2.9 times, 11.8% of steps), reserving synthesis for critical decision points. ADAMEM-HIGH updates more aggressively (33.9%), while ADAMEM-MFT— trained exclusively on instances where dynamic synthesis was decisive — learns a stronger propensity to trigger updates, increasing frequency from 34% to 36% on ALFWorld and from 63% to 74% on WebShop. This behavioral shift explains the higher token consumption of ADAMEM-MFT relative to ADAMEM-HIGH.

*Figure 7.* Example of a successful trajectory produced by ADAMEM-LOW agent. For this ALFWorld task, the agent recovers from both information misalignment and physical constraints through step-wise strategy memory updates.

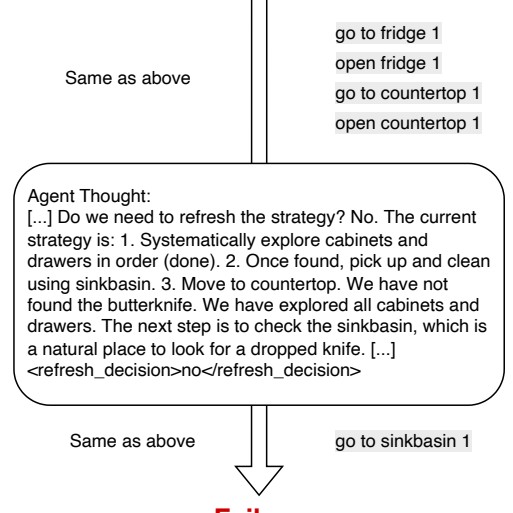

**Task: put a clean butterknife in countertop**

| Strategy Memory | Action |
|---|---|

1. Systematically explore cabinets and drawers in order (starting from cabinet 1) to locate the butterknife, opening closed containers as needed.

go to cabinet 1
open cabinet 1
...
go to cabinet 13
open cabinet 13
go to drawer 1
open drawer 1
go to drawer 2
open drawer 2

2. Once found, pick up the butterknife and clean it using the sinkbasin.

go to cabinet 14
open cabinet 14
...
go to cabinet 17
open cabinet 17

3. After cleaning, move the butterknife to a countertop (e.g., countertop 1, 2, 3, or 4) to fulfill the goal.

Agent Thought:
[...] I have already completed the systematic exploration of cabinets 1-17 and drawers 1-7. The next logical step would be to continue exploring other containers in the environment that might contain the butterknife. [...] I do not believe a strategy refresh is necessary because: The current strategy is systematic and aligned with the task. The failure to find the butterknife so far is due to incomplete exploration, not a misalignment in the strategy. [...]
<refresh_decision>no</refresh_decision>

Same as above

go to fridge 1
open fridge 1
go to countertop 1
open countertop 1

Agent Thought:
[...] Do we need to refresh the strategy? No. The current strategy is: 1. Systematically explore cabinets and drawers in order (done). 2. Once found, pick up and clean using sinkbasin. 3. Move to countertop. We have not found the butterknife. We have explored all cabinets and drawers. The next step is to check the sinkbasin, which is a natural place to look for a dropped knife. [...]
<refresh_decision>no</refresh_decision>

Same as above          go to sinkbasin 1

**Failure**

*Figure 8.* Failure case study on ALFWorld. The agent attempts the task "put a clean butterknife in countertop". After an exhaustive but unsuccessful search of all cabinets and drawers, the agent suffers from *strategy inertia*: it incorrectly assesses that the current strategy remains valid and that the failure is merely due to incomplete exploration. Consequently, it declines to refresh the strategy ($d_{\text{refresh}} = \text{no}$), failing to retrieve necessary external guidance and ultimately failing the task.

*Table 6.* Average strategy update frequency per task on ALFWorld (Qwen3-4B-Instruct, on-policy).

| | Avg. Updates Per Task | Update Frequency (%) |
|---|---|---|
| ADAMEM-LOW | 2.9 | 11.8% |
| ADAMEM-HIGH | 12.0 | 33.9% |

## E. Limitations

**Strategy Inertia.** ADAMEM-LOW relies on prompt-based self-evaluation to decide when to refresh its strategy. This can lead to *strategy inertia*, where the agent rationalizes a failing plan as merely incomplete exploration rather than recognizing that a fundamentally different strategy is required. In such cases, the agent declines to trigger a refresh ($d_{\text{refresh}} = \text{no}$) and continues to follow an obsolete strategy, ultimately exhausting the step budget without solving the task. A concrete example is analyzed in Appendix C.3. Mitigating this failure mode is a promising direction for future work, e.g., by augmenting STEP-MFT with negative examples where stale strategies caused failures, training the model to better detect plan-environment misalignment.

**Successful-Trajectory-Only Memory.** The long-term memory $\mathcal{M}$ is populated exclusively with successful trajectories. Failure trajectories are excluded because in-context learning struggles to derive consistent, actionable insights from them compared to successful demonstrations — agents tend to over-generalize failure patterns or extract conflicting guidance. Developing principled methods for leveraging failure trajectories in the memory bank remains an open challenge.

