# OpenReview forum: "AdaMEM: Test-Time Adaptive Memory for Language Agents"
_ICML.cc/2026/Conference — ICML 2026 regular_

### Official Review · Reviewer_Htnt · 2026-03-09

**Soundness:** 2
**Presentation:** 3
**Significance:** 3
**Originality:** 2
**Overall Recommendation:** 4
**Confidence:** 4

**Summary:**

This paper introduces Adaptive Memory Agent (AdaMEM), a framework for agent test-time adaptation. Briefly, AdaMEM dynamically synthesizes context-aware strategies by conditioning on relevant agent memory and the agent’s current state. Furthermore, the paper introduces Step-MFT, a step-wise memory fine-tuning technique to train agents to generate high-quality strategies from retrieved agent memory. Experiments on ALFWorld and WebShop demonstrate the effectiveness of AdaMEM.

**Compliance With Llm Reviewing Policy:**

Affirmed.

**Final Justification:**

The rebuttal provides additional clarifications and addresses some of the initial concerns on long-horizon tasks. In light of the rebuttal, I have raised my score to a weak accept.

**Key Questions For Authors:**

1. Qwen3-Embedding-4B is used for all retrieval operations. Did the authors try ablating on different embedding models to study retriever’s impact on performance?

2. While the paper explicitly mentions that "all compared memories are constructed based on identical pool of raw successful trajectories”. Can the authors clarify how memories are extracted for ReasoningBank baseline?

**Limitations:**

The paper lacks a discussion on limitations. Could the authors comment on some potential failure modes or any shortcomings of the proposed framework?

**Strengths And Weaknesses:**

**Strengths**
1. The paper is well written and easy to follow. The presented framework, AdaMEM, tackles a relevant and timely problem.

2. Clear motivation. The paper addresses shortcomings of existing approaches by moving away from static retrieval to step-wise test time adaptation, where context-aware strategies are injected dynamically to steer the agent towards the correct solution.

3. Experiments using on-policy and off-policy settings across ALFWorld and Webshop demonstrate AdaMEM variants effectiveness.

**Weaknesses**

1. Equation 1 introduces strategy advantage A(s, z) and defines it as the improvement in value between the memory conditioned and the baseline policy. The dual-filter policy is Section 3.2.2 is a weak heuristic proxy of strategy advantage and does not estimate this advantage. Specifically, outcome success along with action change does not necessarily imply improved success probability since the baseline action might also succeed.

2. Proposition 3.1 also assumes that actions from successful trajectories are optimal. This assumption does not hold for several long horizon tasks (e.g., SWE style tasks), where a successful trajectory can contain several redundant exploration steps.

3. In long horizon tasks, a strategy/retrieved memory can influence future actions and not the immediate action. Strategies that improve downstream decisions but do not affect the immediate action may therefore be discarded by the dual-filter policy, potentially limiting the diversity and usefulness of the curated strategy set.

4. Missing experiments on longer horizon tasks (e.g., a subset of SWE-Bench-Verified). Directly comparing against reported experiments in ReasoningBank using same evaluation setup will make it easier to assess the effectiveness of the proposed method. AdaMEM uses open source models (Qwen and Gemma variants) for experiments, while baselines like ReasoningBank experiment with proprietary LLMs via APIs.

---

> ### Author Rebuttal · Authors · 2026-03-31
>
> ---
> ### **Re: Weakness 1 (Heuristic Proxy)**
>
> Directly estimating strategy advantage $A(s, z)$ via Monte Carlo rollouts or process reward models is **computationally prohibitive** in long-horizon tasks. Step-MFT provides a computation-free proxy for Strategy Advantage, achieving fine-grained credit assignment without auxiliary models.
>
> While this dual-filter *prioritizes high precision over recall* by potentially discarding instances where the baseline might also succeed, it ensures the curated set contains only strategies that actively drive successful decision-making. Empirically, this signal is critical, yielding a +5.3 point absolute gain on ALFWorld after fine-tuning.
>
> ---
> ### **Re: Weakness 2 (Optimality Assumption)**
>
> Proposition 3.1 does not strictly require absolute trajectory optimality. We will revise the text to clarify that the proposition fundamentally requires only a step-level correctness label. In this work, we operationalize this by using sparse outcome rewards as a proxy for process rewards, treating the resulting actions as sufficiently effective rather than strictly optimal.
>
> Empirically, this simple, computation-free approach already demonstrates that minimal process-level signals significantly outperform training-free and outcome-based methods.
>
> Furthermore, the Step-MFT technique is modular. Future optimizations could integrate an LLM-Judge to provide step-level correctness labels [1,2] instead of relying solely on sparse, final rewards.
>
> [1] Xi, Zhiheng, et al. "AgentPRM: Process reward models for LLM agents via step-wise promise and progress." _arXiv preprint arXiv:2511.08325_ (2025).
>
> [2] Choudhury, Sanjiban. "Process reward models for LLM agents: Practical framework and directions." _arXiv preprint arXiv:2502.10325_ (2025).
>
> ---
> ### **Re: Weakness 3 (Strategy Influence)**
>
> AdaMEM-MFT is trained using the AdaMEM-High scaffold, where strategies are designed to be *transient* and limited to the immediate next step. This ensures guidance remains strictly on-policy and tailored to the current state. While estimating long-term strategy influence is a promising future direction, it is unnecessary for the optimization of AdaMEM-high.
>
> ---
> ### **Re: Weakness 4 (Limited Benchmarks)**
>
> We respectfully clarify that ALFWorld and WebShop are considered as standard environments widely used in **long-horizon** agent RL research [3,4]. We exactly follow the configuration of previous work to set the maximum step limit as 50 for ALFWorld and 15 for WebShop. These horizons provide ample complexity to demonstrate the core advantage of our adaptive memory framework.
>
> Due to the limited time during the rebuttal period, we are unable to provide a rigorous evaluation on more *super*-long-horizon benchmarks. We plan to include the results on SWE-Bench in the revision.
>
> [3] He, Shuo, et al. "Hierarchy-of-Groups Policy Optimization for Long-Horizon Agentic Tasks." _ICLR 2026_.
>
> [4] Feng, Lang, et al. "Group-in-Group Policy Optimization for LLM Agent Training." _NeurIPS 2025_.
>
> ---
> ### **Re: Question 1**
>
> *Did the authors try ablating on different embedding models to study retriever’s impact on performance?*
>
> Our primary contribution is the hybrid memory architecture rather than the retrieval component itself. Qwen3-Embedding-4B provides reasonably good quality retrieval for our text-based agent tasks. We will include a comparative study with alternative embedding models in the revision to further characterize the retriever's impact.
>
> ---
> ### **Re: Question 2**
>
> *Can the authors clarify how memories are extracted for ReasoningBank baseline?*
>
> We use the same backbone policy and the original extraction prompts from the ReasoningBank paper. Strategies are distilled offline from the identical pool of successful trajectories used across all compared methods. Embeddings are then computed for these static strategies to enable retrieval at episode initiation. This controlled setup ensures that performance differences reflect architectural design rather than discrepancies in data access.
>
> ---
> ### **Re: Limitation**
>
> *Could the authors comment on some potential failure modes or any shortcomings of the proposed framework?*
>
> 1. AdaMEM-low relies on prompt-based self-evaluation to trigger refreshes, which can result in strategy inertia (Line 417-420 and Appendix C.3). In these cases, agents may insist on following a flawed strategy instead of generating a new one.
>
> 2. Additionally, we restricted retrieval to the top-1 experience to avoid context window overflow. This constraint prevents the agent from synthesizing strategies across multiple retrieved trajectories, which is often required to resolve conflicting evidence.
>
> 3. Finally, we only retrieve successful trajectories. Our early investigation shows that in-context learning struggles to derive actionable insights from failure trajectories compared to successful demonstrations.

---

> > ### Author Rebuttal · Reviewer_Htnt · 2026-04-03
> >
> > Thank you for the detailed rebuttal and clarifications. The authors are encouraged to include additional studies on SWE-Bench, a comparative study on different embedding models and their effect on retriever's performance, and also discuss the above limitations explicitly in the final revision. I will raise my score to a weak accept.

---

### Official Review · Reviewer_JvHj · 2026-03-12

**Soundness:** 3
**Presentation:** 4
**Significance:** 3
**Originality:** 3
**Overall Recommendation:** 5
**Confidence:** 3

**Summary:**

A paper suggests a new approach for agentic memory: instead of using static memory, the authors propose decoupling of long-term memory with trajectories from prior experience and an adaptive short-term memory strategy. The proposed approach, ADAMEM, can be adapted to high and low adaptation efforts for balancing computation cost vs quality ratio. To be able to train an agent to curate long-term memory and strategy memory, the authors suggested the STEP-MFT procedure that involves collecting trajectories obtained with ADAMEM-High inference, dual filtering based on outcome and next action, and RFT. Experiments show superior quality of ADAMEM compared to existing baselines, and the approach establishes a superior Pareto frontier. Carefully designed ablations demonstrated the importance of all the components involved in ADAMEM architecture.

**Compliance With Llm Reviewing Policy:**

Affirmed.

**Final Justification:**

Authors provided answers to some of the raised questions and conducted additional analysis, but still some of questions left without clarification. Overall, I keep my current assessment of the work.

**Key Questions For Authors:**

- How can strategy inertia be addressed? How can ADAMEM address the exploration-exploitation trade-off? Does ADAMEM-High demonstrate the same behaviour?
- Is it necessary to assume optimality of the trajectory in Proposition 3.1?
- Why exactly does ADAMEM-MFT consume more tokens than ADAMEM-High, considering it was trained on data generated on ADAMEM-High?
- It would be interesting to see the number of consumed tokens not per step, but per task. Now it is unclear what overall budget was used by ADAMEM-High and ADAMEM-Low. How many times strategy (Z) is updated in High and Low setups?
- Which state embeddings did you use for the retrieval of long-term memory?
- Why is it necessary to re-generate strategies? Why would it not be sufficient to re-retrieve long-term memory for better decision-making during test time?

**Limitations:**

Limitations are not discussed. Possible limitations that can be considered are hallucination and overconfidence of the agents, as LLMs were used as an agentic backbone.

**Strengths And Weaknesses:**

Strengths:

- Careful experiment design, clear delivery
- Strong performance of ADAMEM agents on the proposed benchmarks compared to the baselines
- Suggested two approaches with different adaptation efforts that allow practitioners to balance the adaptation vs computational cost trade-off.

Weaknesses:

- Providing that ADAMEM agents can suffer from strategy inertia, this effect was not addressed at all. It would be beneficial to indicate how training and data collection procedures can be modified for the elimination of agentic inertia.
- Proposition 3.1 “Necessity of Action Change” suggests that actions from successful trajectories are optimal. However, this is not fully true as the generated strategy based on prior experience in a novel environment can vary from an optimal strategy based on current observations, steering the agent in a maybe successful, but not optimal direction. The claim is too strong; clarification is required.

---

> ### Author Rebuttal · Authors · 2026-03-31
>
> ### **Re: Weakness 1 and Question 1 (Strategy Inertia)**
>
> *How can strategy inertia be addressed?*
>
> To mitigate strategy inertia we can extend Step-MFT with negative strategy examples where outdated plans led to failure. This trains the refresh decision to detect misalignment between environmental feedback and the current strategy. Furthermore, triggering a mandatory refresh after a fixed number of  steps would provide a fallback solution.
>
> *How can ADAMEM address the exploration-exploitation trade-off?*
>
> AdaMEM manages the exploration and exploitation trade off via the refresh decision: maintaining the current strategy acts as exploitation of a known path, while triggering an update drives exploration for alternative approaches when progress stalls.
>
> *Does ADAMEM-High demonstrate the same behaviour?*
>
> AdaMEM-High avoids this inertia entirely. By generating *transient* strategies for immediate use, it continuously forces exploration and re-evaluation, inherently preventing reliance on obsolete plans.
>
> ---
>
> ### **Re: Weakness 2 and Question 2 (Optimality Assumption)**
>
> *Is it necessary to assume optimality of the trajectory in Proposition 3.1?*
>
> Proposition 3.1 does not strictly require absolute trajectory optimality. We will revise the text to clarify that the proposition fundamentally requires only a step-level correctness label. In this work, we operationalize this by using sparse outcome rewards as a proxy for process rewards, treating the resulting actions as sufficiently effective rather than strictly optimal.
>
> Our goal is to demonstrate that this minimal, computation-free proxy significantly outperforms training-free and outcome-based methods. Crucially, the STEP-MFT framework is highly modular. Future optimizations can directly replace our simple heuristics with more advanced techniques for further gains. For example, an LLM-Judge can provide explicit step-level correctness labels [1, 2].
>
> [1] Xi, Zhiheng, et al. "AgentPRM: Process reward models for LLM agents via step-wise promise and progress." arXiv preprint arXiv:2511.08325 (2025).
>
> [2] Choudhury, Sanjiban. "Process reward models for LLM agents: Practical framework and directions." arXiv preprint arXiv:2502.10325 (2025).
>
> ---
>
> ### **Re: Question 3**
>
> *Why does ADAMEM-MFT consume more tokens than ADAMEM-High?*
>
> We hypothesize that AdaMEM-MFT consumes more tokens than AdaMEM-High because the fine-tuning process likely reinforces longer generations. Consequently, the model may learn a correlation between extensive reasoning and task success, prompting it to generate longer reasoning for strategies.
>
> ---
>
> ### **Re: Question 4**
>
> *What is the number of consumed tokens per task?*
>
> As shown in the table below, AdaMEM-low maximizes efficiency, outperforming the strongest baseline (Synapse) while consuming 23% fewer total tokens per task. AdaMEM-max forces strategy update at every step, thus eliminating the tokens spent by AdaMEM-high on when to trigger the update. AdaMEM-max delivers further performance gains with more inference-time compute.
>
> | Method | Tokens Per Task | Success Rate |
> | :--- | :---: | :---: |
> | No Memory | 77K | 45.2 |
> | ReasoningBank | 81K | 49.3 |
> | Synapse | 117K | 52.1 |
> | **AdaMEM-low** | **90K** | **54.0** |
> | **AdaMEM-high** | **150K** | **54.5** |
> | **AdaMEM-max** | **165K** | **65.5** |
>
> *How many times strategy (Z) is updated in High and Low setups?*
>
> As seen from the table below, AdaMEM-high updates the strategy much more frequently as AdaMEM-low. AdaMEM-max updates the strategy at every step.
>
> | Metric | AdaMEM-low | AdaMEM-high | AdaMEM-max |
> | :--- | :---: | :---: |:---: |
> | # Avg strategy update times per task | 2.9 | 12.0 | 27.4 |
> | % Avg strategy update frequency per task | 11.8% | 33.9% | 100%|
>
> ---
> ### **Re: Question 5**
>
> *Which state embeddings did you use for the retrieval of long-term memory?*
>
> Our state embeddings for AdaMEM encode the task description and the actions and observations obtained so far, providing a comprehensive representation of current state.
>
> ---
>
> ### **Re: Question 6**
>
> *Why is it necessary to re-generate strategies? Why would it not be sufficient to re-retrieve long-term memory for better decision-making during test time?*
>
> Re-generating strategies provide a concise summary of key insights from the long-term memory to aid decision-making. Our ablation study in Table 2 ("w/o strategy memory" row) actually shows that re-retrieve long-term memory underperforms our method (59.3 vs 65.5).
>
> ---
>
>
> ### **Re: Limitations**
>
> Please see our response on "Re: Limitations" to Reviewer Htnt.

---

> > ### Author Rebuttal · Reviewer_JvHj · 2026-04-03
> >
> > Thanks to the authors for providing answers and conducting additional analysis.
> >
> > However, some of my questions remain partially unresolved:
> >
> > Question 3.
> > Thank you for explanation. However, your hypothesis wasn't validated on the analysis on correlation of longer generations and success ratio. Your paper would benefit from addressing this.
> >
> > Question 5.
> > It is clear from the paper what embeddings encode. However, it is not clear how they were obtained, which configuration was used (e.g. vector dimension), etc. It is important for reproducibility.

---

> > > ### Author Response · Authors · 2026-04-08
> > >
> > > Thank you for your follow-up questions!
> > >
> > > **Re: Question 3**
> > >
> > > *Why does ADAMEM-MFT consume more tokens than ADAMEM-High?*
> > >
> > > Our analysis suggests that the increased token consumption in AdaMEM-MFT is driven by a higher frequency of strategy updates. Because Step-MFT fine-tunes the model exclusively on instances where dynamic strategy synthesis was necessary for task success, the policy learns a stronger baseline propensity to trigger the token-heavy memory generation ($d_{mem}=yes$) mechanism. Our empirical data confirms this behavioral shift: after fine-tuning, the strategy update frequency increases from 63% to 74% on WebShop and from 34% to 36% on ALFWorld. This naturally drives up the average token cost per step, and we will update the manuscript with this data-backed explanation.
> > >
> > > **Re: Question 5**
> > >
> > > *Which state embeddings did you use for the retrieval of long-term memory?*
> > >
> > > We use an embedding dimension of 2560 (full Qwen3-Embedding-4B output). We build an HNSW index for each memory type in cosine space over L2-normalized embeddings, using `M=32` (graph connectivity), `ef_construction=200` (build-time recall/quality), and `ef_search=100` (query-time recall/speed trade-off). We will release our code with the exact configurations to ensure reproducibility.

---

### Official Review · Reviewer_Rupb · 2026-03-13

**Soundness:** 3
**Presentation:** 3
**Significance:** 3
**Originality:** 3
**Overall Recommendation:** 4
**Confidence:** 3

**Summary:**

This paper proposes ADAMEM, a memory framework for language agents that replaces one-shot, episode-initial memory retrieval with dynamic, step-wise strategy synthesis. The system keeps a long-term memory of successful raw trajectories collected offline, but at test time it generates a short-term natural-language strategy conditioned on the current state, either refreshing it persistently when needed (ADAMEM-LOW) or regenerating it transiently at each critical step (ADAMEM-HIGH). The paper also introduces STEP-MFT, a fine-tuning procedure that keeps only successful training examples where the generated strategy changes the next action, with the goal of teaching the model to produce more decision-relevant strategies. Empirically, the method is evaluated on ALFWorld and WebShop, where it improves over no-memory and static-memory baselines。

**Compliance With Llm Reviewing Policy:**

Affirmed.

**Final Justification:**

The rebuttal addressed my concerns so I maintain my positive score.

**Key Questions For Authors:**

See Weakness.

**Limitations:**

Yes.

**Strengths And Weaknesses:**

**Strength:**

The paper addresses a real and timely problem: current agent-memory systems are often too rigid because they retrieve once at episode start and then rely on stale guidance for the remainder of a long-horizon task. The proposed decomposition into long-term raw trajectory storage and short-term on-the-fly strategy synthesis is intuitive and well motivated, and the distinction between ADAMEM-LOW and ADAMEM-HIGH gives the framework a practical compute/adaptivity knob rather than a single fixed design. The empirical results are also meaningful: on-policy ADAMEM improves over no-memory and static-memory baselines on ALFWorld and WebShop, and the paper includes an off-policy memory-sharing setting plus an ablation showing that both the long-term trajectory memory and the short-term strategy abstraction matter. These are genuine strengths in soundness and significance.



**Weakness:**

(1) The core filtering rule is based on the claim that action change is a necessary condition for strictly positive strategy advantage. But the proposition only establishes a necessary condition under quite strong assumptions: deterministic greedy decoding and the assumption that the action taken in a successful trajectory is optimal. That is much weaker than showing that the retained samples are truly high-advantage or causally useful. In practice, the filter is implemented as an exact lexical mismatch between two action strings, which may be a noisy proxy for meaningful behavioral change.

(2) The paper evaluates on only two benchmarks, with one primary backbone per benchmark, retrieves only top-1 memory, and compares mainly against no-memory and static-memory baselines.

---

> ### Author Rebuttal · Authors · 2026-03-31
>
> ### **Re: Weakness 1 (Strong Assumptions)**
>
> We thank the reviewer for the feedback and would like to provide clarifications below. The core filtering rule does not strictly require that the action taken in a successful trajectory is optimal. We will revise the text to clarify that the framework fundamentally requires only a step-level correctness label. In this work, we operationalize this by using sparse outcome rewards as a proxy, treating actions from successful trajectories as sufficiently effective.
>
> Regarding the lexical mismatch and deterministic decoding, we intentionally utilized lightweight heuristics to validate the core method without heavy computational overhead. Despite being a simple proxy, this approach empirically outperforms training-free and outcome-based filtering baselines.
>
> The Step-MFT framework is highly modular. The core design can seamlessly integrate advanced techniques for future gains. Deterministic decoding can be extended to aggregation over multiple sampled actions. The exact match filter can be upgraded to an LLM-based semantic equivalency detection. Furthermore, an LLM-Judge can replace sparse outcome rewards to provide precise step-level correctness labels [1,2].
>
> [1] Xi, Zhiheng, et al. "AgentPRM: Process reward models for LLM agents via step-wise promise and progress." arXiv preprint arXiv:2511.08325 (2025).
>
> [2] Choudhury, Sanjiban. "Process reward models for LLM agents: Practical framework and directions." arXiv preprint arXiv:2502.10325 (2025).
>
> ---
>
> ### **Re: Weakness 2 (Evaluation Scope)**
>
> ALFWorld and WebShop are widely adopted as rigorous standards to provide sufficient results for long-horizon agentic tasks [3,4]. We evaluate two backbone models (Qwen and Gemma) as the policies for each benchmark.
>
> Additionally, we restrict retrieval to the top-1 experience to avoid context window overflow. This constraint prevents the agent from synthesizing strategies across multiple trajectories, which is often required to resolve conflicting evidence.
>
> Finally, we compare against ReasoningBank and Synapse because they represent the strongest state-of-the-art baselines in agentic memory.
>
> [3] He, Shuo, et al. "Hierarchy-of-Groups Policy Optimization for Long-Horizon Agentic Tasks." _ICLR 2026_.
>
> [4] Feng, Lang, et al. "Group-in-Group Policy Optimization for LLM Agent Training." _NeurIPS 2025_.

---

> > ### Author Rebuttal · Reviewer_Rupb · 2026-04-05
> >
> > Thanks for your response. My concern is resolved and I will maintain my positive score.

---

### Official Review · Reviewer_mRtE · 2026-03-13

**Soundness:** 3
**Presentation:** 3
**Significance:** 2
**Originality:** 3
**Overall Recommendation:** 3
**Confidence:** 3

**Summary:**

The paper presents ADAMEM, which proposes a method for adaptive memory usage during test-time inference in reinforcement learning agents. The framework is designed to dynamically generate strategies for decision-making by combining both long-term trajectory memory and short-term strategy memory. The authors introduce two modes of memory operation (ADAMEM-HIGH and ADAMEM-LOW) to balance the trade-off between adaptability and computational efficiency. Furthermore, the paper introduces STEP-MFT, a fine-tuning technique that improves the model's memory utilization by filtering out redundant strategies. The authors evaluate ADAMEM on two agent benchmarks, ALFWorld and WebShop, showing improvements over baseline models like No Memory, Synapse, and ReasoningBank in certain experimental settings.

**Compliance With Llm Reviewing Policy:**

Affirmed.

**Final Justification:**

My concerns are not fully addressed, so I will keep my score.

**Key Questions For Authors:**

1. How do you expect your method to perform in more complex, high-dimensional environments? Are there any plans for scaling your approach to large-scale RL tasks or multi-agent settings?

**Limitations:**

See weakness

**Strengths And Weaknesses:**

**Strengths and Weaknesses**

**Strengths**

1. The paper presents a dual-memory approach that combines long-term and short-term memory, which could contribute to advancing the way agents handle dynamic task environments and adapt to changing scenarios.

2. The proposed STEP-MFT technique, which filters strategies based on their impact on decision-making, adds a layer of refinement that could potentially be useful in optimizing agent performance for specific tasks.

3. The authors present solid experimental results on ALFWorld and WebShop, demonstrating improvements in certain scenarios. The results are carefully benchmarked against existing memory-based models and reinforce the effectiveness of ADAMEM in specific cases.

**Weaknesses**

1. While the paper introduces an interesting approach to adaptive memory, the improvements in performance over existing methods are marginal. The contributions do not significantly advance the state-of-the-art in the field, especially compared to RL-based methods that show more substantial gains in similar tasks.

2. The authors should provide more detailed analysis on the additional inference cost introduced by the memory design. Understanding this trade-off between the performance gain and the computational cost is crucial to evaluate whether the benefits are worth the increased resource requirements, especially for real-time applications.

3. The results are limited to specific benchmarks, and there is little discussion on how well this method would scale to larger, more complex environments. The evaluation tasks used in the paper, such as ALFWorld and WebShop, are relatively simple and short-horizon in nature. These tasks might not fully demonstrate the advantages of the memory design. The authors should consider testing on more long-horizon tasks, such as project coding or super deep research, where adaptive memory would likely have a more significant impact and showcase the true potential of the method.

---

> ### Author Rebuttal · Authors · 2026-03-31
>
> ---
> ### **Re: Weakness 1 (Performance Gains and Comparison to RL)**
> We thank the reviewer for the feedback but respectfully argue that RL-based methods are NOT the appropriate comparison for this work.
> **AdaMEM focuses on** ***non-parametric test-time adaptation***. Unlike RL-based training-time methods, it requires no expensive parameter updates or specialized critics. It enables agents to self-evolve post-deployment *without* any training.
>
> **Our improvements are far from marginal.** Reviewer JvHj  explicitly recognizes the  ***"strong performance"***  of our framework. For example:
> - On the challenging ALFWorld unseen split, AdaMEM-low surpasses the best existing method (Synapse) by **+6.0** points *without* any training (Line 323-327).
> - For scenarios where compute is less constrained, our AdaMEM-max variant achieves a success rate of **65.5%** (Table 2) compared to the 52.1% by the best existing baseline.
>
> ---
>
> ### **Re: Weakness 2 (Efficiency Analysis and Inference Cost)**
>
> We have already provided detailed efficiency analysis in Appendix D and Table 3, referred by Line 379-383 of the main text. We attach Table 3 below for your reference.
>
> **Table 3: Inference Latency and Performance on ALFWorld (Qwen3-4B)**
>
> | Method | Memory Adaptation (s) | Total Time Per Step (s) | Success Rate (%) |
> | :--- | :---: | :---: | :---: |
> | No Memory | - | 28.5 | 45.2 |
> | ReasoningBank | - | 29.8 (+5%) | 49.3 (+9%) |
> | Synapse | - | 42.7 (+50%) | 52.1 (+15%) |
> | **AdaMEM** | **5.7** | **36.8 (+29%)** | **54.0 (+19%)** |
>
> AdaMEM reduces total inference time by 16% relative compared to Synapse while achieving a higher success rate. This is because unlike trajectory-based retrieval that floods the context window with raw logs, AdaMEM condenses experiences into **concise strategies**, drastically reducing the processing load per step.
>
> If you have any detailed analysis in your mind, please let us know and we are happy to provide additional analysis.
>
> ---
>
> ### **Re: Weakness 3 (Limited Benchmarks)**
>
> We respectfully clarify that ALFWorld and WebShop are considered as standard environments widely used in **long-horizon** agent RL research [1,2]. We exactly follow the configuration of previous work to set the maximum step limit as 50 for ALFWorld and 15 for WebShop. These horizons provide ample complexity to demonstrate the core advantage of our adaptive memory framework.
>
> Due to the limited time during the rebuttal period, we are unable to provide a rigorous evaluation on more *super*-long-horizon benchmarks. We plan to include the results on SWE-Bench in the revision.
>
> [1] He, Shuo, et al. "Hierarchy-of-Groups Policy Optimization for Long-Horizon Agentic Tasks." _ICLR 2026_.
>
> [2] Feng, Lang, et al. "Group-in-Group Policy Optimization for LLM Agent Training." _NeurIPS 2025_.
>
> ---
>
> ### **Re: Question 1**
>
> **Complex, High-Dimensional Environments:**  AdaMEM's decoupled architecture provides a distinct structural advantage. It compresses heavy, multimodal trajectory logs into concise strategies. This prevents context window overflow while retaining actionable guidance.
>
> **Large-Scale RL:**  The framework naturally extends into Hierarchical RL. The dynamically synthesized strategy acts as a high-level sub-goal or reward shaping mechanism. This guides a localized, lower-level RL policy through complex exploration spaces without requiring full sequence rollouts.
>
> **Multi-Agent Settings:**  AdaMEM enables a highly scalable communication protocol. Agents can share isolated, short-term strategies instead of broadcasting massive raw experiential logs. This allows for bandwidth-efficient, decentralized coordination across multiple agents.

---

### Decision · Program_Chairs · 2026-04-30

**Decision:**

Accept (regular)

**Comment:**

This paper proposes a new method to adaptively use the memory in the test time for LLM agents. This framework strategically uses long-term and short-term memories. The framework features two different modes which further strengthens its applicability in real scenarios. Empirical comparision are provided on two standard benchmarks and commendable performance improvements are observed. Particularly, I appreicate that the authors do provide error bars in their results, which is not very common for LLM papers. The results can be further strengthend if more runs are provided. Most concerns from the reviewers are addressed and 3 out of 4 reviewers are positive. I, therefore, recommend Accept.

PS: mRtE does not sufficiently engage in the review process (no rebuttal acknowledgement and a very short final justification) so I downweighted their score after reading their reviews and the author rebuttal myself.